# Gastric Glomus Tumor with Neuroendocrine Features: A Diagnostic Pitfall for Neuroendocrine Tumors

**DOI:** 10.3390/diagnostics15222865

**Published:** 2025-11-12

**Authors:** Dae Hyun Song, Tae-Han Kim, Hyo Jung An

**Affiliations:** 1Department of Pathology, Gyeongsang National University Changwon Hospital, Changwon 51472, Republic of Korea; golgy@hanmail.net; 2Department of Pathology, Gyeongsang National University School of Medicine, Jinju 52727, Republic of Korea; 3Institute of Medical Science, Gyeongsang National University, Jinju 52828, Republic of Korea; 4Department of Surgery, Seoul National University Bundang Hospital, Seongnam 13620, Republic of Korea; taehan.email@gmail.com

**Keywords:** gastric submucosal tumor, glomus tumor, neuroendocrine differentiation, smooth muscle actin

## Abstract

A 60-year-old woman with hypertension and hyperlipidemia was referred for an incidentally detected gastric subepithelial mass during screening endoscopy. Esophagogastroduodenoscopy revealed a 10 mm dimple in the antrum, and contrast-enhanced CT showed a 2.5 cm enhancing oval lesion. Laparoscopic partial gastrectomy with intraoperative endoscopic guidance was performed. Gross examination revealed a 3.0 × 2.0 × 1.0 cm pale, firm nodule. Histology showed small round cells arranged in nests and trabeculae within the muscularis propria, with numerous vessels and focal calcification. Immunohistochemistry was negative for CD117, HMB45, and chromogranin A, but demonstrated strong smooth muscle actin positivity, weak synaptophysin reactivity, and focal CD56 staining. The findings confirmed a gastric glomus tumor with neuroendocrine features. Smooth muscle actin immunostaining is essential to distinguish gastric glomus tumors from neuroendocrine tumors when biopsy material is limited, ensuring accurate diagnosis and appropriate management.

**Figure 1 diagnostics-15-02865-f001:**
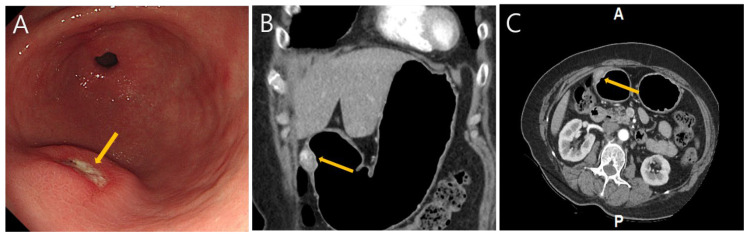
A 60-year-old female patient with BMI 28.5 who had a history of hypertension and hyperlipidemia was referred to regional tertiary medical center for an incidental gastric subepithelial mass diagnosed during national gastric cancer screening esophagogastroduodenoscopy (EGD). She had no diet-related symptoms or signs. The physical examinations were unremarkable and there were no laboratory abnormalities, including anemia or vitamin deficits. EGD showed a single 10 mm crater dimple with an elevated surface located at the anterior surface of the greater curvature of the gastric antrum (**A**). A computerized tomography (CT) scan with gastric protocol was performed; 8 h of fasting was followed by 10 mg of scopolamine (Buscopan; Boehringer Ingelheim, Seoul, Republic of Korea) IV administration and oral administration of gas-producing granules (Robas granules, 4 g/pack; Dong In Dang). The lesion showed a single 2.5 cm oval shaped nodule with increased enhancement located at the distal part of the antrum. There are no notable associations of prominent vascular structure or other abnormalities. ((**B**): coronal image, (**C**): axial image.) Laparoscopic partial gastrectomy with intraoperative endoscopy was performed; the 3 × 2 × 1 cm nodule was excised with clear margins, and the patient was discharged uneventfully on day 6. At the 1-year follow-up, repeat biopsy of the gastric antrum demonstrated mild chronic gastritis without any remarkable pathologic change.

**Figure 2 diagnostics-15-02865-f002:**
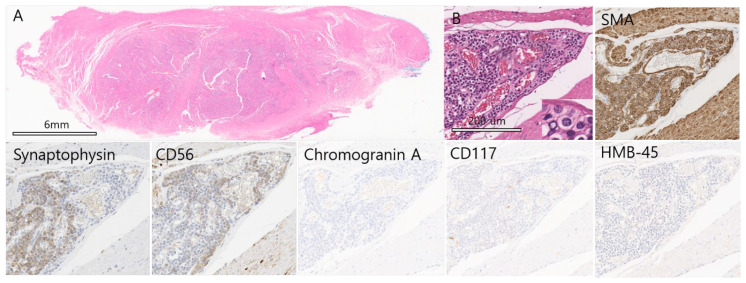
On microscopic examination, Hematoxylin and Eosin staining shows tumor cells infiltrating between bundles of the muscularis propria, forming large and small islands with peripheral trabecular arrangements that could mimic a neuroendocrine tumor (NET). Numerous interspersed blood vessels and occasional calcified nodules are present. No necrosis is seen, and mitotic figures are absent (<1/50 high-power fields) (×10, H&E). (**A**) Tumor cells have moderate cytoplasm and relatively distinct cell borders. Nuclei are uniformly round with finely dispersed chromatin, inconspicuous nucleoli, and regular spacing. (×200, H&E). Inset: tumor cells with round nuclei and well-defined borders (×400, H&E). (**B**) Immunohistochemistry shows diffuse strong positivity for smooth muscle actin (×200). We observed that 80% of central and peripheral tumor cells were weakly positive for Synaptophysin (×200). Fewer than 5% of peripheral tumor cells were weakly positive for CD56 nest (×200, CD56). Staining for Chromogranin A, CD117, and HMB-45 was negative. Gastric glomus tumors are uncommon mesenchymal neoplasms, representing about 1% of gastric mesenchymal tumors [1,2]. They typically present as submucosal masses and may closely mimic gastric neuroendocrine tumor (NET) or gastrointestinal stromal tumors. Small biopsy specimens can be misleading because glomus tumors often show small round cells forming trabecular nests, a pattern that overlaps with NET [3,4,5,6]. Immunohistochemistry is therefore critical. Synaptophysin, a standard NET marker, is positive in roughly 20% of gastric glomus tumors [7], occasionally leading to a provisional NET diagnosis when only limited small tissue is sampled [3]. We report a 60-year-old woman with a 2.5 cm antral subepithelial lesion resected laparoscopically. Histology revealed uniform round cells within the muscularis propria and abundant vasculature. Immunostaining showed strong diffuse smooth muscle actin (SMA) positivity, weak synaptophysin expression in ~80% of tumor cells, focal CD56 positivity, and negativity for CD117 and chromogranin A—features diagnostic of gastric glomus tumor. A review of 16 English-language cases of synaptophysin-positive gastric glomus tumors (including this case) shows that most are present in the antrum, affect patients aged 38–73 years, and consistently express SMA [8,9,10,11]. CD56 is rarely reported and typically weak or peripheral. Because NETs more often arise in the gastric body or fundus [6], antral location and strong SMA positivity are key differentiators for diagnosing gastric glomus tumors. Pathologists should consider SMA staining when a gastric submucosal biopsy shows round cells with well-defined borders and weak synaptophysin or peripheral CD56 expression to avoid misdiagnosis and unnecessary treatment for malignancy.

## Data Availability

The data presented in this article are available upon request from the corresponding author. The data are not publicly available due to privacy.

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
