# Peer review of "Gastric Glomus Tumor with Neuroendocrine Features: A Diagnostic Pitfall for Neuroendocrine Tumors"

_diagnostics, 2025, doi:10.3390/diagnostics15222865_

Round 1
Reviewer 1 Report
Comments and Suggestions for Authors
There is only one coronal CT image (which should be in the venous phase, though it is not specified); it might be proper to add one or more additional CT images in another plane (perhaps axial) or in different phases (arterial for example, if available).
Otherwise, the article seems complete in every respect.
Author Response
Response to the Reviewer’s Comments and Explanation
(Point-by-Point)
Reviewer 1
There is only one coronal CT image (which should be in the venous phase, though it is not specified); it might be proper to add one or more additional CT images in another plane (perhaps axial) or in different phases (arterial for example, if available).
Otherwise, the article seems complete in every respect.
Response: We added an axial image of the APCT. Thank you for your heartful comment.
Reviewer 2 Report
Comments and Suggestions for Authors
- Please show some low-magnification histology images or gross examination pictures to illustrate the anatomical location of this tumor more directly.
- In Figure 2, the authors showed the key positive images for differential diagnosis. If possible, it’s recommended that the authors provide the full IHC panel including negative images important for ruling out gastric NET and GIST, such as CgA, CD56, CD117, DOG-1, and CD34.
- It is recommended that the names of IHC markers be placed on the images directly.
- If available, please provide some follow-up information about this patient to make the case report more complete.
Author Response
Response to the Reviewer’s Comments and Explanation
(Point-by-Point)
Reviewer 2
2. Please show some low-magnification histology images or gross examination pictures to illustrate the anatomical location of this tumor more directly.
Response: We added a low-magnification histology image. Thank you.
2. In Figure 2, the authors showed the key positive images for differential diagnosis. If possible, it’s recommended that the authors provide the full IHC panel including negative images important for ruling out gastric NET and GIST, such as CgA, CD56, CD117, DOG-1, and CD34.
Response: We added a full IHC panel including Synaptophysin, CD56, and HMB-45 for the differential diagnosis. Thank you for your heartful comments.
3. It is recommended that the names of IHC markers be placed on the images directly.
Response: We put the names of the IHC markers on the images directly. Thank you.
4. If available, please provide some follow-up information about this patient to make the case report more complete.
Response: We added one-year follow-up information for this patient. Thank you for your comment.
Round 2
Reviewer 2 Report
Comments and Suggestions for Authors
The authors have made the corresponding revisions and the manuscript is recommended for publication.